# Chromatin Regulator SPEN/SHARP in X Inactivation and Disease

**DOI:** 10.3390/cancers13071665

**Published:** 2021-04-01

**Authors:** Benedetto Daniele Giaimo, Teresa Robert-Finestra, Franz Oswald, Joost Gribnau, Tilman Borggrefe

**Affiliations:** 1Institute of Biochemistry, University of Giessen, Friedrichstrasse 24, 35392 Giessen, Germany; 2Department of Developmental Biology, Erasmus MC, Oncode Institute, Wytemaweg 80, 3015 CN Rotterdam, The Netherlands; t.robertfinestra@erasmusmc.nl (T.R.-F.); j.gribnau@erasmusmc.nl (J.G.); 3Center for Internal Medicine, Department of Internal Medicine I, University Medical Center Ulm, Albert-Einstein-Allee 23, 89081 Ulm, Germany; franz.oswald@uni-ulm.de

**Keywords:** XCI, SHARP, Spen, NCoR, HDAC, polycomb, DNA methylation, transcription, silencing, repression

## Abstract

**Simple Summary:**

Carcinogenesis is a multistep process involving not only the activation of oncogenes and disabling tumor suppressor genes, but also epigenetic modulation of gene expression. X chromosome inactivation (XCI) is a paradigm to study heterochromatin formation and maintenance. The double dosage of X chromosomal genes in female mammals is incompatible with early development. XCI is an excellent model system for understanding the establishment of facultative heterochromatin initiated by the expression of a 17,000 nt long non-coding RNA, known as *X*
*inactive*
*specific*
*transcript* (*Xist*), on the X chromosome. This review focuses on the molecular mechanisms of how epigenetic modulators act in a step-wise manner to establish facultative heterochromatin, and we put these in the context of cancer biology and disease. An in depth understanding of XCI will allow a better characterization of particular types of cancer and hopefully facilitate the development of novel epigenetic therapies.

**Abstract:**

Enzymes, such as histone methyltransferases and demethylases, histone acetyltransferases and deacetylases, and DNA methyltransferases are known as epigenetic modifiers that are often implicated in tumorigenesis and disease. One of the best-studied chromatin-based mechanism is X chromosome inactivation (XCI), a process that establishes facultative heterochromatin on only one X chromosome in females and establishes the right dosage of gene expression. The specificity factor for this process is the long non-coding RNA *X*
*inactive*
*specific*
*transcript* (*Xist*), which is upregulated from one X chromosome in female cells. Subsequently, *Xist* is bound by the corepressor SHARP/SPEN, recruiting and/or activating histone deacetylases (HDACs), leading to the loss of active chromatin marks such as H3K27ac. In addition, polycomb complexes PRC1 and PRC2 establish wide-spread accumulation of H3K27me3 and H2AK119ub1 chromatin marks. The lack of active marks and establishment of repressive marks set the stage for DNA methyltransferases (DNMTs) to stably silence the X chromosome. Here, we will review the recent advances in understanding the molecular mechanisms of how heterochromatin formation is established and put this into the context of carcinogenesis and disease.

## 1. Long Non-Coding RNAs and Cancer

Less than 2% of the genome is transcribed in protein-encoding mRNAs; however, most of it is actively transcribed, which suggests that a fraction produces non-coding RNAs (ncRNAs). ncRNAs are classified based on their size in small ncRNAs (<200 bp) and long ncRNAs (>200 bp, also referred to as lncRNAs) [1,2]. In this review, we focus on lncRNAs.

lncRNAs can be classified based on their genomic localization [3] as well as on their cellular distribution [4]. It is proposed that lncRNAs are organized in secondary and tertiary structures [5] that may offer binding surfaces for proteins containing RNA-recognition motives (RRMs). lncRNAs are capable of interacting with coactivators or corepressors of transcription, recruiting them to specific genes or genomic regions [6,7,8,9]. In addition, lncRNAs are also able to regulate alternative splicing events by interacting with splicing factors [6,10].

Several lncRNAs have been associated with variety of diseases. *Metastasis-associated lung adenocarcinoma transcript 1* (*MALAT1*) was found to be upregulated in renal cell carcinoma (RCC), gastric cancer (GC), gallbladder cancer (GBC), colorectal cancer (CRC), multiple myeloma, clear cell renal cell carcinoma (ccRCC), and glioma, as well as in osteosarcoma [11,12,13,14,15,16,17,18], and it has been proposed as a molecular marker therein [14,15,16,19]. The lncRNA imprinted *H19* gene is maternally expressed and strongly downregulated directly after birth [20,21,22]. It was shown that *H19* is strongly upregulated in gastric cancer [23,24,25], similarly to several other lncRNAs, such as *PVT1* oncogene (*PVT1*), *gastric carcinoma high expressed transcript 1* (*GHET1*), *antisense ncRNA in the INK4 locus* (*ANRIL*), *SPRY4 intronic transcript 1* (*SPRY4-IT1*), and the already mentioned *MALAT1* [18,26,27,28,29,30]. *H19* is also upregulated in other cancer types, such as esophageal cancer, CRC and lung cancer [25]. Another example is represented by *homeobox* (*HOX*) *transcript antisense RNA* (*HOTAIR*), which is upregulated in hepatocellular carcinoma [31], in colorectal cancer [32], in gastric cancer [33], and pancreatic cancer [34].

In this review, we will focus our attention on *X inactive specific transcript* (*Xist*; *XIST* in human), a lncRNA whose main function is to inactivate one X chromosome in female cells to achieve dosage compensation between males (XY) and females (XX) (see below). Recent studies highlighted its frequent deregulation in cancer. *XIST* is responsible for silencing several genes, and the observation that the X-linked oncogenes ARAF-1 and ETS-like 1 (ELK-1) are overexpressed in tumors with multiple active X chromosomes [35] suggests that the deregulation of *XIST* may be associated with cancer. Several studies observed defective X chromosome inactivation (XCI) in breast and basal-like cancer and linked the deregulation of the X chromosome to breast cancer (BC) [36,37,38,39,40,41,42], to ovarian cancer [43], as well as to cancers in patients affected by Klinefelter syndrome [44]. This deregulation is usually given by a loss of *XIST* as result of disappearance of the inactive X chromosome (Xi) and amplification of the active one (Xa) [37,38,40,43,44].

The gathered knowledge of these studies suggest that lncRNAs are important mediators of pathological conditions and they may, in the future, serve as potential therapeutic targets.

XCI serves as a powerful paradigm to study chromatin dynamics at a chromosomal scale. XCI co-evolved with the mammalian sex chromosomes as a mechanism to equalize the dosage of X-encoded genes between male XY and female XX cells. The central player in this process is *Xist*, which was discovered as the first functional lncRNA in mammals, being upregulated from the future Xi, coating the Xi in cis, thereby recruiting chromatin remodelers directly and indirectly rendering the X chromosome inactive. *Xist* is located on the X chromosome and it is surrounded by several other lncRNA-encoding genes, including *Tsix*, *just proximal to Xist* (*Jpx*), and *five prime to XistT* (*Ftx*), which, in mouse, have been shown to be involved in *Xist* regulation through different mechanisms, including transcriptional interference, RNA-mediated recruitment of chromatin remodelers, and through transcription co-activation [45,46,47,48]. *Xist* encodes a 17 kb lncRNA (19 kb in human) that contains six repeat structures that play a crucial, sometimes redundant, role in *Xist*-mediated silencing as well as localization [49]. So far, most of the functional studies have been performed in mouse where deletions of the most 5′ located repeat A led to a silencing phenotype despite the fact that *Xist* spreading was unaffected. Several studies indicated that SHARP [SMRT (silencing mediator for retinoid or thyroid hormone receptors) and HDACs (histone deacetylases)-associated repressor protein], encoded by the *SPEN* (*split ends*) gene [also called *SHARP* or *Mint* (*Msx2-interacting nuclear target protein*)], is a crucial factor in the X inactivation process through interacting with the A repeat sequence and recruitment of several repressor complex members, such as nuclear receptor corepressor (NCoR), SMRT, and nucleosome remodeling deacetylase (NuRD) complexes [50,51,52,53,54,55] (see Table 1).

SHARP is transiently enriched at the promoters and enhancers of genes that are subject to XCI and it recruits NCoR/SMRT complexes that contain HDACs, leading to histone deacetylation [55]. SHARP localization also shows overlap with NuRD complex members predominantly at promoters, and its action is only required during the initiation phase of XCI, as removal of SHARP after Xi is established has no effect [55,140]). As a consequence of the action of SHARP and its associated protein complexes, promoters and enhancers are deacetylated in a stepwise manner, paving the way for the action of the polycomb group (PcG) protein repressive complexes PRC1 and PRC2 that play a crucial role in the establishment and maintenance of the silent state of the Xi. PRC1 is a large multi-protein complex that is recruited to *Xist* through heterogeneous nuclear ribonucleoprotein K (hnRNPK) that acts as a bridge between PRC1 and *Xist* Repeat B and, to a lesser extent, Repeat C [65,66,77]. PRC1-directed deposition of monoubiquitination of K119 of histone H2A (H2A119ub1) is mediated by the core PRC1 complex member really interesting new gene 1 isoform A or B (RING1A/B) and, in turn, is recognized by PRC2 subunit jumanji and AT-rich interaction domain-containing 2 (JARID2) facilitating trimethylation on K27 of histone H3 (H3K27me3) by the enhancer of zeste 2/lysine (K) methyltransferase 6A (EZH2/KMT6A) [141,142,143]. Subsequently, PRC1 and PRC2 recruitment is re-enforced through the recruitment of PRC1 that recognizes the trimethylation of K27 of histone H3 (H3K27me3) through chromobox-containing protein (CBX), which further promotes H2AK119ub1 deposition, facilitating the spreading of silencing [144,145,146]. At a later stage of the XCI process, *de novo* DNA methyltransferases (DNMTs) are recruited to lock in the silent state through the deposition of DNA methylation at promoters and CpG islands (CGI). These studies highlight the concerted action of chromatin readers and writers directing the right order of epigenetic events required to establish the Xi that is propagated through a near infinite number of cell divisions.

## 2. The Inactive X Chromosome Status in Cancer

The complete loss or alteration of the Xi is frequently observed in breast and ovarian cancers, amongst other types of cancer [147,148]. Initial studies showed that *Xist/XIST* RNA is essential for the initiation and establishment of XCI during development, but dispensable to maintain the Xi in female somatic cells [149,150]. Even so, more recent studies making use of more sensitive techniques detect the reactivation of X-linked genes upon nearly complete or partial *Xist/XIST* depletion. The human X chromosome codes for more than 900 coding genes [151], including several tumor suppressor genes and oncogenes [152,153]. Thus, gene dosage changes that are caused by potential reactivation or silencing of X-linked genes could be detrimental. So far, only one well documented study in mice revealed a clear causal relationship between *Xist* deletion in the hematopoietic lineage and high penetrance hematopoietic cancer [154].

In human, the absence of the Xi (Barr body) in female cancer cells and presence of multiple Xa’s have been frequently associated with different forms of cancer, such as breast cancer [38,40,44]. However, these events are primarily attributed to the loss of the Xi and duplication of the Xa due to chromosome segregation errors (see Figure 1) [38,40,44].

Epigenetic alterations that are caused by epigenetic erosion of the Xi have also been described. These erosion events affect histone modification, deposition, and DNA methylation, leading to the reactivation of X-linked genes in breast cancer cell lines and primary tumors [155]. Moreover, the Xi in female cancer genomes has been shown to accumulate more mutations than the autosomes in various cancer types, including medulloblastoma, breast cancer, glioblastoma, and acute myeloid leukemia (AML) [156]. Interestingly, recent studies suggest that high *XIST* expression levels correlate with a poor survival in various types of cancer [157]. Some of these studies propose that *XIST* acts as a competing endogenous RNA (ceRNA) [158,159], by depleting microRNAs. As a consequence, specific RNA targets cannot be degraded, which may lead to the dysregulation of downstream genes [160,161]. So far, both epigenetic and genetic changes have been observed in relation to the Xi of cancer cells, but whether these alterations are driving events that give a selective advantage to cancer cells is under debate. Nevertheless, evidence suggests that the Xi epigenetic status and *XIST* expression levels are potential cancer biomarkers as a readout for genomic instability or epigenomic changes. Therefore, understanding the factors and mechanisms that render and maintain the X chromosome inactive, both during embryonic development and in somatic cells during the maintenance phase of XCI, is of crucial importance.

## 3. Chromatin Modifiers That Act in XCI

The regulation of the X chromosome is controlled by chromatin modifiers that build up heterochromatin formation by deacetylating and methylating histone tails, finally leading to the DNA methylation of regulatory CpG islands (see Figure 2).

Specific enzymes that play a central role in XCI are HDACs, the PRC1 and PRC2 complexes, and DNMTs (see Table 1). Recently, the SHARP protein has been identified as a direct *Xist* interactor. This protein bridges *Xist* to HDACs allowing for histone deacetylation at the X chromosome. This section discusses the current knowledge about SHARP and other key chromatin modifiers that are involved in XCI.

### 3.1. SHARP

SHARP is a protein of more than 400 kDa that contains four RRMs and a highly conserved C-terminal domain, called SPOC (Spen paralog and ortholog C-terminal domain), which is responsible for mediating the repressive function of SHARP [162]. There is a family of SPOC domain-containing proteins that includes RNA binding motif protein 15/one-twenty-two (RBM15/OTT1) and RNA binding motif protein 15B/one-twenty-two protein 3 (RBM15B/OTT3) [163,164,165], which have recently been linked to XCI [52,166,167].

The SHARP-encoding gene was originally identified by Newberry and colleagues while screening an expression library from mouse brain to identify novel interaction partners for the Homeo domain transcriptional repressor Msx2 (Homolog of Muscle Segment Homeobox 2, Msh Homeobox 2) using a Farwestern approach. They named the interacting protein MINT. The full length protein was reported to have 3576 amino acids and three RRMs within the amino-terminal part [168]. Because 68 missing residues in the amino-terminal part of the original MINT protein analysis, they did not identify RRM1 in this report [168]. Subsequently, Shi and colleagues performed a yeast two hybrid screen using a mouse whole embryo E17 library and the carboxy-terminus of NCoR2 (also known as SMRT). Sequence information from the mouse clone was used to screen a human cDNA library. The full length human cDNA coded for a 3651 amino acid protein, which they named SHARP. The SMRT interacting protein fragment that was identified by Shi et al. corresponded to the C-terminus of SHARP, which they named Repression Domain (RD, now referred to as SPOC) [50]. Interestingly, the cDNA clone encoding for the MSX2-interacting protein fragment that was isolated by Newberry et al. [168] corresponds to amino acids 2138 to 2462 in MINT (Q62504.2), and it is closely related to the later reported receptor interaction domain (RID) in the human MINT homolog SHARP [50]. To search for novel components of the Notch signaling pathway, we also performed a yeast two hybrid screen using a human embryonic brain library and the DNA binding transcription factor (TF) of Notch signaling, recognition signal binding protein for immunoglobulin kappa J region [RBPJ; also known as CSL (CBF1, Suppressor of Hairless, Lag-1)], as a bait. This screen identified a cDNA encoding for a protein identical to SHARP. SHARP (NP_055816.2) consists of 3664 residues and the RRM1 was identified at the very amino-terminus [169]. The highly conserved RBPJ interaction domain (RBPID) of SHARP was fine mapped from residues 2882 (2776 in MINT) to 2839 (2814 in MINT) and structure information of the RBP-J-SHARP/MINT complex became available meanwhile [170]. Interestingly, BLAST analysis identifies 79% identity between the human and mouse SHARP proteins.

In regard to the in vivo function of SHARP, its mouse homolog, MINT, has been studied, making use of knockout models. *Mint* knockouts were primarily analyzed for its function in the Notch signal transduction pathway, since SHARP is a pivotal cofactor at Notch target genes [162,169,170,171,172,173,174,175]. *Mint* knockout mice are embryonic lethal at around day E12.5-14.5 and they show, amongst others, cardiac and pancreatic defects, as well as an increased number of marginal zone B cells [176]. Whether there is a difference in lethality between male and female embryos was not studied. Further studies making use of *Mint* knockout mice unveiled the role of *Mint* in the thymus supporting early T-cell development [177]. Additional studies have described the function of MINT, in regulating the expression of the osteocalcin-encoding gene [168,178] and *collagen type II alpha 1 chain* (*Col2a1* [179]).

Early studies described SHARP as a regulator of nuclear receptors-dependent transcription by recruiting the HDACs-containing corepressor SMRT complex via the SPOC domain [50]. The same study also highlighted the ability of SHARP to bind lncRNAs, such as *steroid receptor coactivator* (*SRA)*, to modulate gene expression [50]. However, SHARP is not an exclusive regulator of nuclear receptors-dependent transcription. In fact, it has also been linked to the highly conserved Notch signaling pathway that is involved in regulating different developmental and differentiation events and it is frequently dysregulated in cancer ([180,181,182], see Table 1). As discussed above, SHARP was recently identified in a screen for Xist RNA binding proteins (see Table 1) and it was shown to be essential for X-inactivation in embryonic stem cells (ESCs) [51,52,53,54,55,129]. However, SHARP does not bind exclusively *SRA* and *Xist* but also retroviral RNAs that are characterized by regions with structural similarity to the A-repeat of *Xist* [128], which is required for the *Xist*/SHARP interaction [51,53,130]. SHARP was shown to bind to the *SRA* lncRNA by its RRMs. The four RRMs of SHARP are located at its amino-terminal portion (aa 1–600, see Figure 3A). Solving the crystal structure of SHARP RRM2, RRM3, and RRM4 (see Figure 3B), Arieti and colleagues could demonstrate that RRM3 and RRM4 form an inter-domain platform (see Figure 3B orange and red), whereas RRM2 is not part of this platform (see Figure 3B, yellow). Additional RNA binding studies showed that the RRM3/RRM4 platform interacts with the H12-H13 region of *SRA,* whereas RRM2 seems not to be involved in this interaction [183]. Moreover, the role of RRM1 (see Figure 3A) in RNA binding has not been elucidated. Structure homology modeling suggests that the highly conserved amino acids 6 to 81 in SHARP form a typical RRM topology (see Figure 3C). In addition, structure alignments of RRMs identify the typical amino acids in essential positions needed for interactions with nucleotides (see Figure 3D–F). Although RRM2 alone seems not to be involved in *SR*A binding, one could speculate that RRM1 and RRM2 also form an intermolecular platform to bind specific lncRNAs. Structure analyses of a SHARP protein, including all four RRMs together with additional RNA binding studies, will give more insights into the exact role of each RRM and potential cooperative effects of RNA binding and regulation of gene transcription. In summary, lncRNA-mediated gene regulation by SHARP needs at least the amino terminal RRMs for RNA binding as well as the carboxy-terminal SPOC domain (see Figure 3A) for the recruitment of epigenetic modifiers.

In this review, we summarize and extract what is known regarding SHARP as a transcription and chromatin regulator, which will be useful for understanding its recently emerged role in XCI.

#### 3.1.1. SHARP in Chromatin Regulation

SHARP has been characterized as the central corepressor at Notch target genes, forming the bridge between the transcription factor RBPJ and several corepressors [181]. In short, RBPJ either interacts with the Notch coactivator or with the SHARP corepressor based on the activation status of the Notch signaling pathway. This repressor-activator switch is carefully controlled by the Notch intracellular domain (NICD) and, consequently, its turnover determines the amplitude and duration of the Notch response [162,180,181,185].

SHARP has been initially characterized as a key component of the RBPJ-associated corepressor complex [169] by functioning as a platform for the recruitment of additional corepressors, such as eight-twenty-one/myeloid translocational gene 8 (ETO/MTG8) and C-terminal binding protein/CtBP-interacting protein (CtBP/CtIP), finally bridging RBPJ to HDACs [172,173,174]. Recent studies have further defined the interactome of the SPOC domain of SHARP (referred to as SPOCome) and shed light on a surprising and unexpected mechanism. In fact, SHARP does not exclusively interact with corepressors via its SPOC domain, but also with coactivators [162,171]. In this study, we observed that SPOC is able to interact in a mutually exclusive fashion with the corepressor NCoR, an ortholog of SMRT (also known as NCoR2) that has been previously linked to the Notch signaling pathway similarly to SMRT [186,187,188,189,190], or with the coactivator lysine (K) methyltransferase 2D (KMT2D) [171]. Given that the KMT2D complex has histone H3K4 methyltransferase activity and that the NCoR complex contains HDACs, this competition allows for balancing acetylation on lysine 27 of histone H3 (H3K27ac) and trimethylation of H3K4 (H3K4me3), fine-tuning the expression of Notch target genes [171]. It must be noted that this NCoR-KMT2D competition is strongly dependent on the phosphorylation status of NcoR, which has been previously shown to be phosphorylated on two highly conserved serine residues within its C-terminal LSDSD motif [191,192,193,194,195]. We observed that phospho-NCoR outcompetes KMT2D for binding to SPOC promoting the repression of Notch target genes [162,171]. In conclusion, this study emphasized that SHARP is more than a simple corepressor, but it operates as a poising factor that balances the repression and activation of Notch target genes. Therefore, SHARP might be a potential therapeutic target for both cancers in which Notch has been defined as a tumor suppressor as well as malignancies in which Notch acts as an oncogene. To reach this goal, thermodynamic studies [196] and the recently characterized crystal structure of the RBPJ/SHARP complex [170] may be useful. In fact, these data may allow developing small molecules modulating the RBPJ/SHARP interaction. These molecules can be further validated in biochemical and functional analysis with the final goal to get clinical relevance as therapeutic drugs. Additionally, it might be possible to develop inhibitors of the SPOC-NCoR/SMRT interaction to block the repressive activity of SHARP and reactivating the Notch pathway in those tumors in which Notch acts as a tumor suppressor. Additionally, in this case, the crystal structure of the SPOC-NCoR/SMRT complex may be instructive [197].

As discussed above, SHARP promotes XCI via the recruitment of deacetylating complexes through its SPOC domain [51,52,53,54,55,129]. It is important to also note that RBM15/OTT1 has been identified as a *Xist* interactor, marking further the importance of SPOC domain-containing proteins in XCI [52,54]. RBM15/OTT1 and its paralog RBM15B/OTT3 bridge *Xist* to the m^6^A methylation machinery promoting m^6^A methylation of *Xist* [167]. In line with that, Wilms tumor 1 (WT1)-associated (WTAP) and VIRILIZER proteins, subunits of the m^6^A methylation machinery have been previously identified as *Xist* interactors [52] and the *Drosophila* homolog of RBM15/OTT1, known as Nito, is also involved in m^6^A RNA methylation [198].

In addition, the SPOC domain of SHARP has been described to interact with the ubiquitin-conjugating enzyme (E2) UbcH8 (also known as UBE2L6 (ubiquitin conjugating enzyme E2 L6)) decreasing Notch-dependent promoter activity in a SHARP-dependent fashion [199]. It has also been suggested that SHARP homodimerizes through its SPOC domain, leading to a reduction of its repressive activity through RBPJ in luciferase assays [200]. Once more, these studies highlight the complex function of SHARP marking the requirement for a better comprehension of its molecular functions.

#### 3.1.2. SHARP Directly Interacts with the NCoR/HDAC Complex

As mentioned above, SHARP was initially identified to interact in a yeast-2-hybrid screen with SMRT [50]. Later on we could show that it interacts with NCoR in a phospho-dependent manner [171]. The NcoR and SMRT complexes are composed of several subunits with different specific functions (see Table 1) and mutations of their components have been observed in cancer, as well as intellectual and developmental disorders (see Table 1). Purification studies of the NCoR and SMRT complexes unveiled their subunits composition identifying the following ones [79,80,201,202,203,204,205]: SMRT or NCoR1 themself; transducin β-like protein 1 (TBL1) whose gene is located on the X chromosome and mutated in human sensorineural deafness [206]; transducing β-like 1 (TBL1)-related protein (TBLR1); the deacetylase HDAC3; and, G-protein pathway suppressor 2 (GPS2), an intracellular signaling protein. Structural studies have also shed light on the interaction between HDAC3 and SMRT [207]. Smrt knockout results in severe heart defects that lead to the death of most of the embryos at day E16.5 [208]. Jepsen and colleagues could overcome this limit using myocyte-specific reexpression of *Smrt* and observed that Smrt is also required for forebrain development [190]. On the other hand, Ncor knockout leads to defects in erythropoiesis, T cell, and neural development [209]. Finally, Hdac3 knockout is embryonic lethal and its conditional knockout in liver results in hepatocellular carcinoma [210,211].

Among all of the subunits of the NCoR and SMRT complexes the HDACs that confer deacetylation activity to those complexes and determine their transcriptional repression ability are probably the most important subunits. However, recent studies have also suggested that HDACs can play a positive role in transcription and this is also true for HDAC3 [212,213,214]. This positive function is given by the fact that HDACs deacetylate histone proteins, but also non histone proteins. We have found that HDAC3 promotes the deacetylation of NICD1 increasing its stability and as consequence its transcriptional activity [212]. It is proposed that HDAC3 is found exclusively within the NCoR and SMRT complexes and required for their catalytic activity [215,216]. However, it still needs to be investigated whether HDAC3 can also work independently of the NCoR and SMRT complexes to fulfill its positive role in gene transcription. In fact, it cannot be excluded that other proteins that are different than SMRT and NCoR may be able to stimulate the catalytic activity of HDAC3.

NCoR and SMRT have been linked to the regulation of several different DNA binding proteins, such as the transcriptional repressor B cell lymphoma 6 (BCL6) [217,218], thyroid hormone receptor (THR) [203], RBPJ [171,186,187,188,189], the oncogenic fusion protein acute myeloid leukemia 1/eight-twenty-one (AML1/ETO) [219], TEL and c-JUN [220], and REV-ERBα, which is a transcription factor that is involved in the circadian clock [221]. SHARP interacts with NCoR and SMRT [171,222], recruiting them to the DNA, as we have briefly described above. Structural studies clarified how this interaction occurs and how it is regulated: it involves arginine (R) 3552 and 3554 of SHARP (within the SPOC domain) and serines (S) 2449 and 2451 of NCoR [162,171,223]. Furthermore, this interaction is dependent on the phosphorylation status of these serine residues within NCoR and at least one of them is phosphorylated by casein kinase 2 (CK2) [171,191,192,193,194,195]. Again, the availability of the crystal structure of the SPOC domain of SHARP in combination with phosphoSMRT [224] may be useful in developing molecules to inhibit this interaction as a potential therapeutic option. In addition, the inhibitors of CK2 may be a powerful tool to prevent the interaction between SHARP and SMRT or NCoR. It is important to note that SHARP is phosphorylated by p21-activated kinase 1 (PAK1) within the SPOC domain at S3486 and at threonine (T) 3568 [225]. PAK1-dependent phosphorylation of SHARP augments its repressive activity in luciferase assays [225]. However, whether this phosphorylation impacts on the SHARP-SMRT/NCoR interaction is unknown, and it would be important to evaluate that with inhibitors of PAK1 to potentially destabilize this interaction.

#### 3.1.3. Pathological Deregulation of SHARP

SHARP has been linked to several diseases both because of mutations that occur within the gene or because of its altered function, localization, and/or expression (see Table 1). Frameshift and non-sense mutations of the *SPEN* gene have been described in adenoid cystic carcinoma [131] and in mantle cell lymphoma (MCL) [132,133,134]. SHARP mutations have been also described in diffuse large B-cell lymphoma (DLBCL) [135], in splenic marginal zone lymphoma (SMZL) [136,137], in pancreatic adenosquamous carcinoma (PASC) [138], as well as in neurodevelopmental disorders (NDDs) [139]. However, whether these mutations deregulate the Notch pathway and/or XCI is not clear. The identification of mutations in other Notch pathway components in the same type of tumor [131,132,133,134,136,137] would suggest that these mutations may have a negative impact on the Notch signaling pathway. However, SHARP does not exclusively regulate XCI and the Notch signaling pathway; in fact, it regulates the estrogen receptor α (ERα)-dependent transcription and mutations of *SHARP* have been detected in breast cancer, where it acts as a tumor suppressor [226] Similarly, SHARP expression is upregulated in colorectal adenocarcinoma, where it is described to deregulate the Wnt signaling pathway [227].

SHARP is mislocalized in myotonic dystrophy [228], while, in acute myeloid leukemia (AML), it has been proposed to have an altered function as consequence of its interaction with the oncofusion protein AML1/ETO deregulating the Notch signaling pathway [173,174]. Following the same line of reasoning, mutations of the genes encoding for SHARP interactors might have a deleterious ending. This might be the case of subunits of the KMT2D complex; in fact, *KMT2D* and *lysine demethylase 6A*/*ubiquitously transcribed tetratricopeptide repeat protein X-linked* (*KDM6A*/*UTX*) mutations have been observed in patients that are affected by Kabuki Syndrome [229,230,231,232,233,234,235,236,237] and, in line with that, *kmt2d* knockout in zebrafish recapitulates the Kabuki phenotype and it is characterized by the deregulation of the Notch pathway [238].

Similarly to SHARP, the SPOC domain-containing RBM15/OTT1 protein has also been linked to diseases. For example, it is translocated in acute megakaryocytic leukemia [239,240] and significantly upregulated in patients with blast-crisis chronic myelogenous leukemia (CML) [241].

These studies further mark the relevance of the proteins containing a SPOC domain and the importance to better characterize the function of the SPOC domain in normal as well as pathological conditions. Hitherto, it remains completely unclear as to whether XCI is affected in the same diseases in which SHARP is dysfunctional.

### 3.2. PRC1 and PRC2

It has been shown that the lncRNA Xist is key for the recruitment of Polycomb complexes to the future Xi [65,77]. The Polycomb group (PcG) genes were originally identified in *Drosophila melanogaster* [242]. Their products are organized in two different multisubunit complexes that are known as PRC1 and PRC2, which are involved in building up a repressive chromatin environment. As previously introduced, PRC1 promotes H2AK119ub1, while PRC2 deposits H3K27me3. Recent studies also elucidated additional PRC complexes that differ from each other based on subunits composition [76].

All of the PRC1 complexes contain RING1A or 1B (see Table 1). The subcomplex type is defined by the PcG ring finger protein (PCGF 1–6). RING and PCGF form a core unit that is common to all the PRC1 complexes, and this unit is associated with additional specific subunits, including polyhomeotic homolog (PHC, isoform 1–3), sex comb on midleg homolog (SCMH, isoform L1 or L2), and one of the chromobox homolog (CBX, either isoform 2, 4, 6, 7, or 8) proteins. Non-canonical PRC1 complexes do not contain CBX, but they still associate with RING1 and YY1 binding protein (RYBP and YAF2) cofactors. In the case of PRC2 complexes, EZH1/2, retinoblastoma binding protein (RBBP, it can be isoform 4 or 7), suppressor of zeste 12 (SUZ12), and embryonic ectoderm development (EED) form the core that associates with different subunits, giving rise to the PRC2.1 or PRC2.2 complexes. To note subunits of both PRC1, PRC2 as well as ncPRC1 have been observed in different types of cancer (see Table 1).

Polycomb complexes follow a specific sequence of events to help silence the future Xi (see Figure 2). First, non-canonical PRC1 is recruited via heterogeneous nuclear ribonucleoprotein K (hnRNPK), which also interacts with *Xist* [65,77]. This complex promotes H2AK119ub1 on the Xi in response to *Xist* expression, and this modification is recognized by the PRC2 complex through its subunit JARID2 [106]. PCR2 establishes H3K27me3 domains due to the activity of EZH1/2 [77]. Subsequently, H3K27me3 is recognized by the canonical PRC1, which enforces the silencing on the X chromosome [144,243]. Other lncRNAs might use similar mechanisms to promote the PRC2-dependent spreading of H3K27me3, for example, Airn and Kcnq1ot1 [244].

### 3.3. DNA Methyltransferases (DNMTs)

XCI is locked in through DNA methylation that occurs on the fifth carbon of cytosine, indicated as 5mC. 5mC usually occurs at regions of the genome that contain the CpG dinucleotide and it is enriched at repetitive sequences as well as within gene bodies. CpG dense regions are usually located near the Transcription Starting Site (TSS) of genes and they are defined as CpG islands or CGI [245]. Usually, the methylation of CGI is associated with gene silencing and it is highly stable. However, recent studies unveiled that hydroxylation of 5mC by ten-eleven translocation (TET) enzymes facilitates the rapid reactivation of silenced target genes [245].

DNA methylation is catalyzed by DNMTs, which are grouped into two main classes: the de novo DNMTs such as DNMT3A and DNMT3B establish the DNA methylation pattern during early embryogenesis, while the maintenance DNMT, DNMT1, restore the DNA methylation pattern after DNA replication [246,247,248]. 5mC marks are subsequently read by proteins that contain dedicated domains and that bridge 5mC to additional enzymes to further support the establishment of repressive chromatin [249]. Three different classes of 5mC readers are known: readers that contain the methyl binding domain (MBD); Kaiso and Kaiso-like proteins that contains the broad complex, tramtrack, and bric a brac/Pox virus and zinc finger (BTB/POZ) domain and Krüppel-like C2H2 zinc fingers; finally, proteins containing the SET and RING finger-associated (SRA) domain [250]. It is also important to note that unmethylated CGI are targets of dedicated proteins, for example, CXXC finger protein 1 (CFP1) [251,252].

The long term maintenance of X chromosome inactivation is achieved via DNA methylation that occurs at promoter-associated CGIs. This methylation is catalyzed by DNMT3B (see Figure 2 and Table 1), and it occurs with two different kinetics at different regions of the X chromosome [56]. At most CGIs, DNA methylation occurs slowly and requires the chromosomal binding of structural maintenance of chromosomes hinge domain-containing 1 (SMCHD1), while, at a small proportion of CGIs, DNA methylation is SMCHD1-independent through a very fast process.

## 4. Conclusions

LncRNAs are key players in many different cellular processes. Here, we have reviewed the recent discoveries how heterochromatin formation, initiated by lncRNA *Xist*, is established. XCI is a tightly regulated process that is controlled by multiple epigenetic regulators, amongst them SHARP, that, in recent years, has been shown to play a central role in the silencing of the future Xi. Mechanistically, it will be interesting to find out whether SHARP’s main function is to recruit the HDAC3-containing NCoR1/2 complexes or whether additional interaction partners will be required for XCI. Furthermore, the function of the N-terminal RRM1 of SHARP, most likely acting in concert with RRM2, remains to be elucidated. Alterations of SHARP and additional XCI-contributing factors have been identified in several cancer types, but to which extend they contribute to cancer progression remains to be determined, as well as whether these alterations affect the Xi status. Studies focusing on male and female specific differences could help to understand their potential therapeutic value.

## Figures and Tables

**Figure 1 cancers-13-01665-f001:**
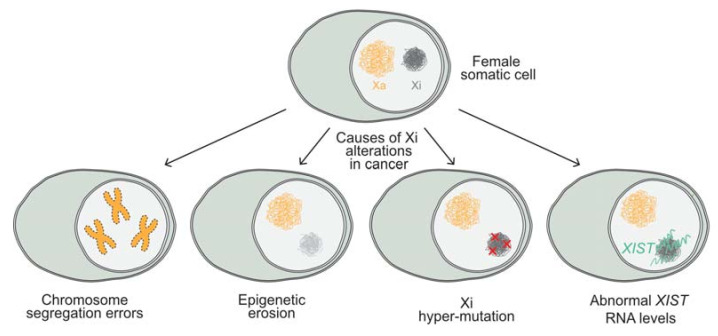
Possible causes of alteration of the Xi in cancer. Female somatic cells have one active and one inactive X chromosome, alterations of the Xi, often observed in cancer. These alterations can be caused by chromosome segregation errors, often leading to loss of the Xi and duplication of the Xa. Epigenetic erosion can lead to reactivation of X-linked genes. Mutations in the Xi happen more often than in other chromosomes. Abnormal *Xist* RNA levels are also observed in cancer. Xa = Active X chromosome, Xi = Inactive X chromosome.

**Figure 2 cancers-13-01665-f002:**
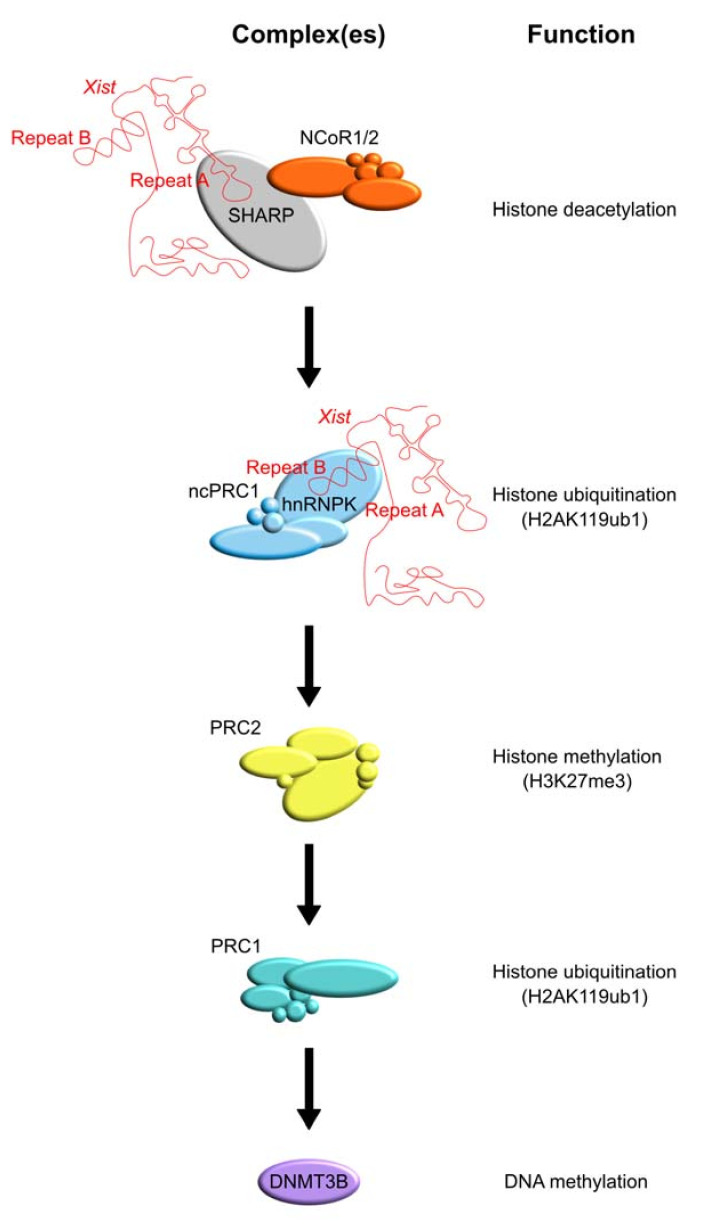
Proposed model for the silencing of the future inactive X chromosome (Xi) in female cells. The lncRNA *Xist* recruits SHARP [SMRT (silencing mediator for retinoid or thyroid hormone receptors) and HDACs (histone deacetylases)-associated repressor protein] to the X chromosome upon initiation of X chromosome inactivation (XCI). On one side, SHARP interacts with *Xist* through its RRMs (RNA recognition motifs) while on the other side it recruits chromatin modifiers through its highly conserved SPOC (Spen paralog and ortholog C-terminal) domain. One of the SPOC interactors is the multisubunit NCoR1/2 (nuclear receptor corepressor) complex that promotes histone deacetylation through its subunit HDAC3. As a next step, *Xist* interacts with hnRNPK (heterogeneous nuclear ribonucleoprotein K) recruiting the non-canonical PRC1 complex (ncPRC1) that writes H2AK119ub1 through RING1A/B (really interesting new gene 1 isoform A/B). Subsequently, H2AK119ub1 is recognized by JARID2 (jumanji and AT-rich interaction domain-containing 2), subunit of the PRC2 complex that writes H3K27me3 through EZH2/KMT6A [enhancer of zeste 2/lysine (K) methyltransferase 6A]. H3K27me3 is read by canonical PRC1 through its subunit CBX (chromobox-containing protein) and H2AK119ub1 is further established on the chromatin. Finally, silencing is achieved due to the activity of DNA methyltransferase 3B (DNMT3B) that methylates position 5 of cytosines (5 mC) within the DNA.

**Figure 3 cancers-13-01665-f003:**
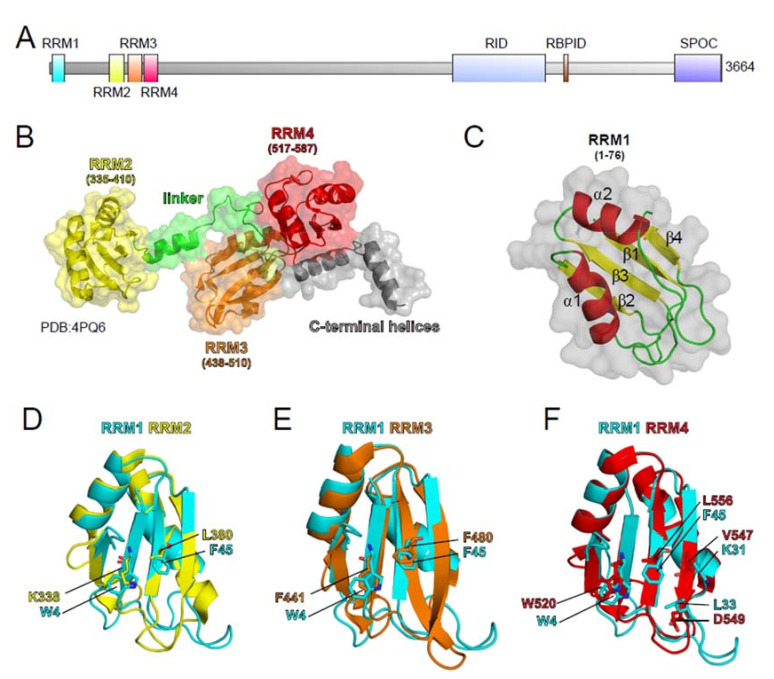
Structure modeling of SHARP RRM1. (**A**) Schematic representation of SHARP protein domains: RRM1 (aa 6 to 81), RRM1 (aa 335 to 410) RRM3 (aa 438 to 510), RRM4 (aa 517-587), Receptor Interaction Domain (RID, aa 2201 to 2707), RBP-J Interaction Domain (RBPID, aa 2776 to 2813), Spen Paralogue and Orthologue C-Terminal Domain (SPOC, aa 3417 to 3664). (**B**) Crystal structure of SHARP RRMs 2 to 4 (aa 335 to 620) from PDB: 4PQ6 [183]. (**C**) Homology model of SHARP (amino acids 6 to 81, accession NP_055816.2) performed with swissmodel (https://swissmodel.expasy.org/ (accessed date 14 January 2021). The model shows the typical β1-α1-β2-β3-α2-β4 topology of RRMs: Four antiparallel β sheets (yellow) framed from two α helices (red). Loops are shown in green. (**D**–**F**) Structural alignment of RRM1 (cyan) with RRM2 (**D**, yellow), RRM3 (**E**, orange), and RRM4 (**F**, red). Amino acids typically having contact with nucleic acids are highlighted. All of the structures and alignments were performed with PyMol [184].

**Table 1 cancers-13-01665-t001:** Proteins and complexes involved in the regulation of X chromosome inactivation (XCI). The “Disease(s)” column indicates diseases caused by mutations in the XCI related genes/proteins described in this table. The functional link between these mutations and XCI remains to be investigated.

Protein/Complex	Subunits	Function(s) in XCI	Disease(s)	References
DNMT3B	-	DNA methyltransferase	AML, FSHD, HD, ICF, PR	[56,57,58,59,60,61,62,63,64]
hnRNPK	-	Bridging protein between *Xist* and ncPRC1	AKS, AML, KLS, KS, MF, OS	[65,66,67,68,69,70,71,72,73,74,75]
ncPRC1	PCGF3/5 RING1A/B RYBP/YAF	E3 ubiquitin ligase	MDS	[65,76,77,78]
NCoR1/2 *	GPS2 HDAC3 NCoR1/2 * TBL1 TBLR1	Deacetylase	ASDs, BC, CC, HCC, ID, MB, NDDs, OMZL, PS, SCZ	[79,80,81,82,83,84,85,86,87,88,89,90,91,92,93,94,95]
PRC1	CBX2/4/6/7/8 PCGF1-6 PHC1-3 RING1A/B SCMH1/L2	E3 ubiquitin ligase/Recognition of histone methylation	BC, DD, DSD, ESCC, GC, MCL, MDS, OSS, PM	[76,78,96,97,98,99,100,101,102,103,104,105]
PRC2	AEBP2 EZH2 ** EED JARID2 RBBP4/7 SUZ12	Methyltransferase	AML, DS-AMKL, DLBCL, ETP-ALL, FL, HCC, MDS, MPN, T-ALL, T-PLL	[76,106,107,108,109,110,111,112,113,114,115,116,117,118,119,120,121,122,123,124,125,126,127]
SHARP	-	Adaptor protein that recruits the HDAC3-containing NCoR1/2 complexes	ACC, BC, DLBCL, MCL, NDDs, PASC, SMZL	[51,52,53,54,55,128,129,130,131,132,133,134,135,136,137,138,139]

ACC: Adenoid cystic carcinoma; AEBP2: Adipocyte enhancer-binding protein 2; AKS: Au–Kline syndrome; AML: Acute myeloid leukemia; ASDs: Autism spectrum disorders; BC: Breast cancer; CC: Colon cancer; CBX2/4/6/7/8: Chromobox homolog 2/4/6/7/8; DD: Developmental disorder; DLBCL: Diffuse large B-cell lymphoma; DNMT3B: DNA methyltransferase 3B; DS-AMKL: Acute megakaryoblastic leukemia associated with Down syndrome; DSD: Disorders of sex development; EED: Embryonic ectoderm development; ESCC: Esophageal squamous cell carcinoma; ETP-ALL: Early T-cell precursor acute lymphoblastic leukaemia Early T-cell precursor acute lymphoblastic leukaemia; EZH2: Enhancer of zeste 2; FL: Follicular lymphoma; FSHD: Facioscapulohumeral dystrophy; GC: Gastric cancer; GPS2: G-protein pathway suppressor 2; HCC: Hepatocellular carcinoma; HD: Hirschsprung disease; HDAC3: Histone deacetylase 3; hnRNPK: Heterogeneous nuclear ribonucleoprotein K; ICF: Immunodeficiency, centromeric instability and facial anomalies; ID: Intellectual disability; JARID2: Jumanji and AT-rich interaction domain-containing 2; KLS: Kabuki-like syndrome; KS: Kabuki syndrome; MB: Medulloblastoma; MCL: Mantle cell lymphoma; MDS: Myelodysplastic syndromes; MF: Mycosis fungoides; MPN: myeloproliferative neoplasm; NCoR1/2: Nuclear receptor corepressor; ncPRC1: non-canonical PRC1 complex; NDDs: Neurodevelopmental disorders; OMZL: Ocular marginal zone lymphoma; OS: Okamoto syndrome; OSS: Osteosarcoma; PASC: Pancreatic adenosquamous carcinoma; PCGF1-6: PcG ring finger 1-6; PCGF3/5: PcG ring finger 3/5; PHC1-3: Polyhomeotic homolog 1-3; PM: Primary microcephaly; PR: Prostate cancer; PRC1: Polycomb repressive complex 1; PRC2: Polycomb repressive complex 2; PS: Pierpont syndrome; RBBP4/7: Retinoblastoma binding protein 4/7; RING1A/B: Really interesting new gene 1A/B; RYBP/YAF: RING1 And YY1 Binding Protein/YY1-associated factor; SCMH1/L2: Sex comb on midleg homolog 1/L2; SCZ: Schizophrenia; SHARP: SMRT (silencing mediator for retinoid or thyroid hormone receptors) and HDACs (histone deacetylases)-associated repressor protein; SMZL: Splenic marginal zone lymphoma; SUZ12: Suppressor of zeste 12; T-ALL: T-cell acute lymphoblastic leukemia; T-PLL: T-cell prolymphocytic leukemia; TBL1: Transducin β-like protein 1; TBLR1: Transducing β-like 1 (TBL1)-related protein; XCI: X chromosome inactivation; *Xist*: *X inactive specific transcript*; * NCoR2 is also known as SMRT (silencing mediator for retinoid or thyroid hormone receptors); ** EZH2 is also known as KMT6A (lysine (K) methyltransferase 6A).

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
