# Peer review of "Chromatin Regulator SPEN/SHARP in X Inactivation and Disease"

_cancers, 2021, doi:10.3390/cancers13071665_

Round 1

Reviewer 1 Report

Major comments

The authors provide an overview of molecular mechanisms of X chromosome inactivation and underline the knowledge gaps in the field. The manuscript mainly focuses on the functions of the long non-coding RNA X inactive specific transcript and its corepressor, SHARP/SPEN. Despite several typing/grammar/style errors (see the minor comments below), this review is well written and easy to follow. A summary figure at the end of the manuscript underlining the gaps in our knowledge of X chromosome inactivation and proposed future research directions would significantly improve the review.

Page 5, line 184-185: Could the authors provide a clearer picture on how SHARP was discovered and how conserved this protein is across mammals?

Page 8, lines 281-287:

The authors provide an interesting avenue for SHARP research towards developing therapeutic drugs.

Can they elaborate more on how exactly thermodynamic studies and the crystal structure of RBPJ/SHARP and SPOC-NCoR/SMRT complexes could help in achieving this goal? Could they propose a more detailed experimental design for this specific purpose?

Minor comments

Page 1, line 18: remove “involves”

Page 2, line 53: should be “capable of interacting”

Page 2, line 56: should be “associated with”

Page 2, line 68: should be “whose” am not “which its”

Page 2, line 71:  should be “responsible for”

Page 3, line 113: should be “by the core”

Page 4, line 144: replace  “, alterations of the Xi are often observed in cancer. These alterations” with “Alterations of the Xi, often observed in cancer,”

Page 4, line 146: remove “that”

Page 4, lines 151-153: this sentence is unclear. Did the author mean “X chromosome” and not ‘Xi”? What did authors mean by “than the other chromosomes”?

Page 4, line 156: replace “implicated to” with “which can”

Page 4, line 167: should be “deacetylating”

Page 5, line 176: should be “As a next step”

Page 6, line 210: replace comma with a full stop after “transcription”

Page 6, lines 210-211: replace “it has been subsequently linked also” with “ it has also been linked”

Page 6, lines: 213: 218: this sentence is too long. Split it in 2-3 sentence for clarity.

Page 7, line 249: provide the reference for PyMol

Page 9, line 313: replace “made” with “shed”

Page 9, lines 329-332: this sentence is too long and should be split in two.

Page 9, line 353: should be “goal of using”

Page 9, line 356: Did the author mean “the SPEN gene” here?

Page 10, line 367: should be “Following the same line of reasoning”

Page 10, line 372: should be “is characterized by the” instead of “present with”

Page 10, line 375: replace comma with a full stop after “diseases”

Page 10, line 385: which chromosome(s) are Polycomb group genes located on?

Page 10, line 389: should be “differ from”

Page 10, line 393: should be “(PHC, isoforms 1-3)

Page 10, line 394: remove “either”

Page 10, line 412: remove “it”

Page 11, line 414-417: split the sentence in two

Page 11, line 418-419: should be “the de novo”

Page 11, lines 438-439: this sentence is unclear

Page 11, lines 439-440: should be “The initial binding of bridging factor SHARP to Xist and the NCor/HDAC complex defines the first step in XCI”

Author Response

Major comments

The authors provide an overview of molecular mechanisms of X chromosome inactivation and underline the knowledge gaps in the field. The manuscript mainly focuses on the functions of the long non-coding RNA X inactive specific transcript and its corepressor, SHARP/SPEN. Despite several typing/grammar/style errors (see the minor comments below), this review is well written and easy to follow. A summary figure at the end of the manuscript underlining the gaps in our knowledge of X chromosome inactivation and proposed future research directions would significantly improve the review.

We want to thank the reviewer for her/his helpful suggestions which allowed us to significantly improve our manuscript.

Page 5, line 184-185: Could the authors provide a clearer picture on how SHARP was discovered and how conserved this protein is across mammals?

As requested by the reviewer, we now describe how SHARP was identified and we discuss possible future directions in the Conclusions section.

Page 8, lines 281-287:

The authors provide an interesting avenue for SHARP research towards developing therapeutic drugs.

Can they elaborate more on how exactly thermodynamic studies and the crystal structure of RBPJ/SHARP and SPOC-NCoR/SMRT complexes could help in achieving this goal? Could they propose a more detailed experimental design for this specific purpose?

We have followed the indications of the reviewer and modified the manuscript accordingly.

Minor comments

Page 1, line 18: remove “involves”

Page 2, line 53: should be “capable of interacting”

Page 2, line 56: should be “associated with”

Page 2, line 68: should be “whose” am not “which its”

Page 2, line 71: should be “responsible for”

Page 3, line 113: should be “by the core”

Page 4, line 144: replace “, alterations of the Xi are often observed in cancer. These alterations” with “Alterations of the Xi, often observed in cancer,”

Page 4, line 146: remove “that”

Page 4, lines 151-153: this sentence is unclear. Did the author mean “X chromosome” and not ‘Xi”? What did authors mean by “than the other chromosomes”?

In the publication from Jäger et al. (PMID: 24139898), the authors observe that the inactive X chromosome (Xi) accumulates more mutations than the autosomes. We have clarified that in the text.

Page 4, line 156: replace “implicated to” with “which can”

Page 4,

Page 5,

Page 6,

Page 6, linked”

line 167: should be “deacetylating”
line 176: should be “As a next step”
line 210: replace comma with a full stop after “transcription”
lines 210-211: replace “it has been subsequently linked also” with “ it has also been

lines: 213: 218: this sentence is too long. Split it in 2-3 sentence for clarity. line 249: provide the reference for PyMol
line 313: replace “made” with “shed”
lines 329-332: this sentence is too long and should be split in two.

Page 6,
Page 7,
Page 9,
Page 9,
Page 9,
Page 9,
Yes, exactly. We indicated all the genes using the italic style.
Page 10, line 367: should be “Following the same line of reasoning”
Page 10, line 372: should be “is characterized by the” instead of “present with” Page 10, line 375: replace comma with a full stop after “diseases”

line 353: should be “goal of using”
line 356: Did the author mean “the SPEN gene” here?

Page 10, line 385: which chromosome(s) are Polycomb group genes located on?

For “Polycomb group (PcG) genes” we indicate all the genes that encode for the different subunits of the PRC1 and PRC2 complexes. These genes are located on several different chromosomes.

Page 10, line 389: should be “differ from”

Page 10, line 393: should be “(PHC, isoforms 1-3)

Page 10, line 394: remove “either”

Page 10, line 412: remove “it”

Page 11, line 414-417: split the sentence in two

Page 11, line 418-419: should be “the de novo”

Page 11, lines 438-439: this sentence is unclear

Page 11, lines 439-440: should be “The initial binding of bridging factor SHARP to Xist and the NCor/HDAC complex defines the first step in XCI”

We are grateful to the reviewer for the detailed suggestions and we have addressed all her/his suggestions accordingly.

Reviewer 2 Report

The review from Giaimo et al focuses on the molecular mechanisms that regulate X inactivation. The review describes in detail the role of several molecular players and will be a useful resource for the scientific community investigating X inactivation. However, there are a few points that should be addressed more in detail, especially related to the interplay between the molecular machinery regulating X inactivation and signalling pathways and their alterations in cancer. In addition, this Reviewer believes that adding a Table summarising the role of the molecular players mentioned and improving the current Figures will facilitate the reader. 

Major points:

1) The authors should add a Table summarising the role of the main epigenetic and molecular players mentioned. 

2) Cancer is mentioned in a superficial ways in several points, especially at the beginning of the review (e.g. line 57 "different types of cancer"; line 139 "different forms of cancer"; line 154 "various types of cancer"). It would be very helpful to add a Paragraph addressing the role of X inactivation in one/multiple cancer types where robust literature is available. Moreover, alterations of the molecular players mentioned in cancer should be added to the Table (see point 1).

3) The link with the Notch signalling is very intriguing. The add of a dedicated paragraph expanding the part related to the interplay between Notch and X inactivation, would increase the novelty of the manuscript. A Figure explaining this would even further highlight the relevance of such interactions. 

4) Figures 1 and 2 are poorly drawn. Figure 1 has a very low resolution, which should be improved. Figure 2 is not really understandable, the authors should redrawn it focusing on the main message they want to deliver. 

5) The Conclusion section is not adequate. What are the main point to be investigated in the future?

Author Response

The review from Giaimo et al focuses on the molecular mechanisms that regulate X inactivation.
The review describes in detail the role of several molecular players and will be a useful resource for the scientific community investigating X inactivation. However, there are a few points that should be addressed more in detail, especially related to the interplay between the molecular machinery regulating X inactivation and signalling pathways and their alterations in cancer. In addition, this Reviewer believes that adding a Table summarising the role of the molecular players mentioned and improving the current Figures will facilitate the reader.

We want to thank the reviewer for the positive evaluation of our review and for her/his constructive suggestions and criticisms.
Major points:
1) The authors should add a Table summarising the role of the main epigenetic and molecular players mentioned.

We added a Table as suggested by the reviewer.

2) Cancer is mentioned in a superficial ways in several points, especially at the beginning of the review (e.g. line 57 "different types of cancer"; line 139 "different forms of cancer"; line 154 "various types of cancer"). It would be very helpful to add a Paragraph addressing the role of X inactivation in one/multiple cancer types where robust literature is available. Moreover, alterations of the molecular players mentioned in cancer should be added to the Table (see point 1).

We want to thank the reviewer for her/his suggestions and we have corrected the text accordingly.
3) The link with the Notch signalling is very intriguing. The add of a dedicated paragraph expanding the part related to the interplay between Notch and X inactivation, would increase the novelty of the manuscript. A Figure explaining this would even further highlight the relevance of such interactions.

Thanks for the suggestion. We would LOVE to do this. However, so far we don’t have any evidence, apart from the shared cofactor SHARP/SPEN. In our view, SHARP/SPEN has a ‘moonlightning’ function, repression of Notch target genes and facilitating XCI.
A possible link between Notch and XCI would suggest binding of RBPJ to or near
Xist or Tsix however, there are no RBPJ binding sites close to Xist or Tsix. At least in some leukemia cell lines, there is no particular pattern observed at the X-chromosome before and after adding Notch inhibitior (Dino, correct in Beko, correct in other examples of T-ALL)?

4) Figures 1 and 2 are poorly drawn. Figure 1 has a very low resolution, which should be improved. Figure 2 is not really understandable, the authors should redrawn it focusing on the main message they want to deliver.
We have corrected the figures as requested by the reviewer.

5) The Conclusion section is not adequate. What are the main point to be investigated in the future?
We have corrected the Conclusion section accordingly.

Reviewer 3 Report

In the present review by Giaimo et al., the authors summarize the latest research data on the X-chromosome inactivation (XCI) phenomenon. This is a complex and important biological process, and disturbance of XCI leads to the development of various pathologies.

Firstly, the authors summarize the role of Long non-coding RNAs and XCI in carcinogenesis. The second major part is devoted to the description of chromatin modifiers that act in X-inactivation processes.

I have some major and minor comments

Major comments

  1. In fact, the review focuses on describing the functions of SHARP protein in XCI as well as in non-XCI processes. From this point of view, the title of the article should be changed to a more specialized one. The current title is very broad and authors do not describe all new aspects and achievements in XCI phenomenon.

In particular, great success in recent years has been achieved in matters of:

-Xist lncRNA in three-dimensional genome architecture and role of the Polycomb proteins in these processes;  

- interactions of Xist with matrix attachment proteins like hnRNP U and CIZ1;

- mechanism of escaping the genes from X inactivation.

  1. When the authors describe the functions of the SHARP protein in normal and pathological conditions, it is really unclear which of these functions are associated with XCI and which ones are not. The authors should clearly separate these two aspects.

Minor comments

  1. Lines 53-55 “lncRNAs are capable to interact with coactivators or corepressors of transcription as well as with splicing factors, thereby recruiting them to specific genes or genomic regions [6-10].”

What is the evidence that lncRNAs not only interact with splicing factors but also recruit them to chromatin?

  1. Lines 71-72 “XIST is responsible of silencing several genes and given that some of them act as oncogenes,” Please, make description: “of several genes”? What exactly do you mean?
  2. Line 83 “Xist which was discovered as the first functional ncRNA gene” Probably “functional lncRNA gene”? First identified non-coding RNA genes were rRNA and tRNA genes.
  3. Line 178 “H2AK199ub1” should be “H2AK119ub1”
  4. Line 199 “Mint knock-out mice are embryonic lethal at around day E12.5-14.5” Are there any known differences in lethality between male and female embryos? If yes, please, indicate it.
  5. Line 272 “acetylation on lysine 4 of histone H3 (H3K27ac)” should be “acetylation on lysine 4 of histone H3 (H3K4ac)”
  6. “HNRNPK” should be written as “hnRNPK”

Author Response

In the present review by Giaimo et al., the authors summarize the latest research data on the Xchromosomeinactivation (XCI) phenomenon. This is a complex and important biological process, and disturbance of XCI leads to the development of various pathologies.

Firstly, the authors summarize the role of Long non-coding RNAs and XCI in carcinogenesis. The second major part is devoted to the description of chromatin modifiers that act in X- inactivation processes.

I have some major and minor comments

We are grateful to the reviewer for her/his suggestions that allowed us to significantly improve our manuscript.

Major comments

1. In fact, the review focuses on describing the functions of SHARP protein in XCI as well as in non-XCI processes. From this point of view, the title of the article should be changed to a more specialized one. The current title is very broad and authors do not describe all new aspects and achievements in XCI phenomenon.

In particular, great success in recent years has been achieved in matters of:

- Xist lncRNA in three-dimensional genome architecture and role of the Polycomb proteins in these processes;

- interactions of Xist with matrix attachment proteins like hnRNP U and CIZ1;

- mechanism of escaping the genes from X inactivation.

We want to thank the reviewer for her/his suggestion. The new title is: “Chromatin regulator SPEN/SHARP in X inactivation and cancer”

2. When the authors describe the functions of the SHARP protein in normal and pathological conditions, it is really unclear which of these functions are associated with XCI and which ones are not. The authors should clearly separate these two aspects.

We want to mark that to our knowledge there are no pathological links between SHARP and XCI probably due to the novelty of the link between SHARP and Xist in XCI. We have modified the text following the indications of the reviewer.

Minor comments

1. Lines 53-55 “lncRNAs are capable to interact with coactivators or corepressors of transcription as well as with splicing factors, thereby recruiting them to specific genes or genomic regions [6-10].”

What is the evidence that lncRNAs not only interact with splicing factors but also recruit them to chromatin?

We apologize for the confusion and we have corrected the text accordingly. Now we wrote:

“lncRNAs are capable to interact with coactivators or corepressors of transcription recruiting them to specific genes or genomic regions [6-9] and in addition, lncRNAs are also able to regulate alternative splicing by interacting with splicing factors [6,10].”

2. Lines 71-72 “XIST is responsible of silencing several genes and given that some of them act as oncogenes,” Please, make description: “of several genes”? What exactly do you mean?
We apologize for the confusion and we have corrected the text. Now we wrote:

“XIST is responsible of silencing several genes and the observation that the X-linked oncogenes ARAF-1 and ETS-like 1 (ELK-1) are overexpressed in tumors with multiple active X chromosomes [35], suggest that deregulation of XIST may be associated with cancer.”

3. Line 83 “Xist which was discovered as the first functional ncRNA gene” Probably “functional lncRNA gene”? First identified non-coding RNA genes were rRNA and tRNA genes.
We have corrected this accordingly.
4. Line 178 “H2AK199ub1” should be “H2AK119ub1”

We have corrected this accordingly.

5. Line 199 “Mint knock-out mice are embryonic lethal at around day E12.5-14.5” Are there any known differences in lethality between male and female embryos? If yes, please, indicate it.

We do not know whether there is a difference in lethality and we have stressed this point in the text. We wrote:

Mint knockout mice are embryonic lethal at around day E12.5-14.5 and show cardiac and pancreatic defects as well as an increased number of marginal zone B cells [93]. However, whether there is a difference in lethality between male and female embryos is not known.”

6. Line 272 “acetylation on lysine 4 of histone H3 (H3K27ac)” should be “acetylation on lysine 4 of histone H3 (H3K4ac)”

We have corrected this accordingly.

7. “HNRNPK” should be written as “hnRNPK”

We have corrected this accordingly.

Round 2

Reviewer 2 Report

The authors addressed the majority of this reviewer's concerns.

Additional points:

1) The authors generated a Table, as suggested by this Reviewer. However, this Reviewer also suggested to include information about alterations of the genes/proteins listed in cancer. This Reviewer still believes that this would be a valuable add to the Table to summarise disease and cancer types where altered X inactivation is observed.

2) The new title does not entirely reflect the content of the manuscript. If the authors intend to keep this title, they should modify at least the Abstract and the Introduction.  

Author Response

Reviewer 2

The authors addressed the majority of this reviewer's concerns.

Additional points:

1) The authors generated a Table, as suggested by this Reviewer. However, this Reviewer also suggested to include information about alterations of the genes/proteins listed in cancer. This Reviewer still believes that this would be a valuable add to the Table to summarise disease and cancer types where altered X inactivation is observed.

2) The new title does not entirely reflect the content of the manuscript. If the authors intend to keep this title, they should modify at least the Abstract and the Introduction.  

We want to thank the reviewer for her/his comments that helped us to further improve our manuscript. We have followed the suggestions of the reviewer and completed the table as requested. Of note, it still remains to be investigated  whether the observed mutations  are linked with X-inactivation. We have also modified the manuscript following the content of the title however, we want to stress that paragraph 3.1.3 covers the pathological function of SHARP as stated in the title.

Reviewer 3 Report

Dear authors,

thank you for your job that signigicant improve the review.
I just detected one mistake again: Fig. 2 “H2AK199ub1” should be “H2AK119ub1”

Author Response

Reviewer 3

Dear authors,

thank you for your job that signigicant improve the review.
I just detected one mistake again: Fig. 2 “H2AK199ub1” should be “H2AK119ub1”

We are grateful to the reviewer for her/his help. We corrected Fig. 2 accordingly.